# Vitamin D Metabolites and Clinical Outcome in Hospitalized COVID-19 Patients

**DOI:** 10.3390/nu13072129

**Published:** 2021-06-22

**Authors:** Sieglinde Zelzer, Florian Prüller, Pero Curcic, Zdenka Sloup, Magdalena Holter, Markus Herrmann, Harald Mangge

**Affiliations:** 1Clinical Institute of Medical and Chemical Laboratory Diagnostics, Medical University of Graz, Auenbruggerplatz 15, 8036 Graz, Austria; sieglinde.zelzer@medunigraz.at (S.Z.); florian.prueller@medunigraz.at (F.P.); pero.curcic@medunigraz.at (P.C.); zdenka.sloup@bbgraz.at (Z.S.); markus.herrmann@medunigraz.at (M.H.); 2Institute for Medical Informatics, Statistics and Documentation, Medical University of Graz, 8036 Graz, Austria; magdalena.holter@medunigraz.at

**Keywords:** vitamin D metabolites, COVID19, clinical outcome

## Abstract

(1) Background: Vitamin D, a well-established regulator of calcium and phosphate metabolism, also has immune-modulatory functions. An uncontrolled immune response and cytokine storm are tightly linked to fatal courses of COVID-19. The present retrospective study aimed to inves-tigate vitamin D status markers and vitamin D degradation products in a mixed cohort of 148 hospitalized COVID-19 patients with various clinical courses of COVID-19. (2) Methods: The serum concentrations of 25(OH)D_3_, 25(OH)D_2_, 24,25(OH)_2_D_3_, and 25,26(OH)_2_D_3_ were determined by a validated liquid-chromatography tandem mass-spectrometry method in leftover serum samples from 148 COVID-19 patients that were admitted to the University Hospital of the Medical Uni-versity of Graz between April and November 2020. Anthropometric and clinical data, as well as outcomes were obtained from the laboratory and hospital information systems. (3) Results: From the 148 patients, 34 (23%) died within 30 days after admission. The frequency of fatal outcomes did not differ between males and females. Non-survivors were significantly older than survivors, had higher peak concentrations of IL-6 and CRP, and required mechanical ventilation more frequently. The serum concentrations of all vitamin D metabolites and the vitamin D metabolite ratio (VMR) did not differ significantly between survivors and non-survivors. Additionally, the need for res-piratory support was unrelated to the serum concentrations of 25(OH)D vitamin D and the two vitamin D catabolites, as well as the VMR. (4) Conclusion: The present results do not support a relevant role of vitamin D for the course and outcome of COVID-19.

## 1. Introduction

Vitamin D deficiency is a highly prevalent condition in developed countries around the globe [1] that can easily be corrected by supplementation. An insufficient supply of vitamin D adversely affects calcium-phosphate metabolism and is associated with an increased risk of osteomalacia and rickets, the two classic manifestations of vitamin D deficiency [2]. In addition to its role in the metabolism of calcified tissues, vitamin D may also have an immune-modulatory function [3], and is discussed in context with several other diseases including cardiovascular and autoimmune disease, multiple sclerosis, and cancer [4], although intervention trials often remain disappointing. The immune modulatory effects of vitamin D have led to speculations that deficiencies of this vitamin may facilitate an excessive immune reaction in COVID-19 patients. Established risk factors for severe COVID-19 include advanced age, obesity, chronic illness, non-Caucasian ethnicity, and work-related contact with COVID-19 patients (6). Furthermore, several studies suggest that genetic factors [5] and vitamin D deficiency [6,7] may also increase the risk for severe COVID-19- and SARS-CoV-2-related mortality.

Vitamin D appears to modulate immune function in several ways. Existing evidence suggests interactions with macrophages, B and T lymphocytes, neutrophils, and dendritic cells [3]. T and B lymphocytes express 1α-hydroxylase, and thus can synthesize 1,25-dihydroxyvitamin D (1,25(OH)_2_D_3_), the active metabolite of vitamin D [8]. In a comprehensive review focusing on the influence of vitamin D on immune function [3], several experimental in vitro studies were discussed. The immuno-modulative facets of 1,25(OH)_2_D_3_ included the ability to contain an excessive immune response by modulation of T-helper (TH)-17 cells [9,10,11]. Other reviews discuss vitamin D as an anti-inflammatory modulator in respiratory tract infections [12,13]. These immune-modulatory effects have sparked speculations that an adequate supply of vitamin D may reduce the risk of respiratory tract infections and facilitate a balanced immune response in SARS-CoV-2-infected patients [14,15,16]. In line with this theory, a recent meta-analysis found that low serum concentrations of 25-hydroxyvitamin D (25(OH)D), the inactive precursor of 1,25(OH)_2_D_3_, are associated with severe infectious disease and mortality [17]. Moreover, vitamin D deficiency seems to increase one’s susceptibility to influenza and other respiratory tract infections [12]. A retrospective study of 2000 critically ill patients showed that serum 25(OH)D concentrations below 75 nmol/L are associated with higher odds for developing a severe respiratory tract infection and acquired respiratory distress syndrome (ARDS) [18]. In line with this result, Jain et al. found a stronger inflammatory response in COVID-19 patients with vitamin D deficiency [19], which may be due to direct antiviral effects of vitamin D against enveloped viruses [20,21]. Supplementation studies provide additional support for the hypothesis that vitamin D ameliorates disease severity of COVID-19 patients [22,23]. However, beside immune-modulatory and anti-viral effects [21], vitamin D also activates the renin-angiotensin-aldosterone system (RAAS) [24], and increases angiotensin converting enzyme-2 receptor (ACE2) expression [25]. As SARS-Cov-2 enters the body by binding to ACE2 receptors in the respiratory tract, excessive vitamin D concentrations may also increase the risk of infection.

While there is some evidence for beneficial effects of vitamin D in SARS-Cov-2-infected patients, other studies challenge this concept [25,26]. Cereda et al. speculated that vitamin D supplementation might even facilitate a severe course of COVID-19 through an overwhelming macrophage response and a subsequent cytokine storm [25]. It is also unclear whether the strong inflammatory response of critically ill SARS-Cov-2 patients decreases the 25(OH)D serum concentration by accelerating vitamin D metabolism and by decreasing hepatic vitamin D-binding protein synthesis, which carries approximately 99% of total 25(OH)D in serum. A major limitation of most existing studies is the measurement of 25(OH)D by immunoassays [27,28], which are known for their variable analytical performance, especially in critically ill patients [27,29]. The advent of liquid-chromatography-tandem-mass spectrometry (LC-MS/MS) allows the accurate quantitation of 25(OH)D and its principal catabolite, 24,25(OH)_2_D_3_. Cavalier et al. has suggested that the parallel analysis of these two metabolites facilitates an assessment of vitamin D metabolism that is based on dynamic physiologic principles. In particular, the proportion of 25(OH)D and 24,25(OH)_2_D_3_ takes the individual set-point for vitamin D sufficiency into account, and may thus detect early functional vitamin deficiencies [30]. The 25,26(OH)_2_D was recently brought into connection with intestinal calcium transport. The accurate measurement of 25,26(OH)_2_D requires a chromatographic separation from 24,25(OH)_2_D, because 24,25(OH)_2_D has the same molecular mass as 25,26(OH)_2_D. This analytical approach is provided by our laboratory [31,32].

The present study aimed to address existing inconsistencies by measuring 25(OH)D, 24,25(OH)_2_D_3_, and 25,26(OH)_2_D_3_ in a mixed cohort of hospitalized COVID-19 patients and to explore their predictive potential for a fatal outcome.

## 2. Material and Methods

### 2.1. Study Design

In this retrospective study, the serum concentrations of 25(OH)D_3_, 25(OH)D_2_, 24,25(OH)_2_D_3_, and 25,26(OH)_2_D_3_ were measured by a validated LC-MS/MS method [33] in leftover blood samples from 148 COVID-19 patients that were admitted to the University Hospital of the Medical University of Graz between April and November 2020. Residual serum samples from the first routine blood collection that were sent to our laboratory for routine diagnosis were used for analysis. After the completion of all routine laboratory testing, residual material was stored at −80 °C until batched analysis. Anthropometric and clinical data, as well as outcome data were obtained from the laboratory and hospital information systems. The primary outcome was death within 30 days after admission. Respiratory support was used as the secondary end-point. The study was approved by the institutional ethics committee of the Medical University of Graz (EK 32-475 ex 19/20).

### 2.2. Laboratory Analysis

The diagnosis of COVID-19 was confirmed in all patients with viral reverse transcriptase PCR using the Xpert^®^ Xpress SARS-CoV-2 (singleplex) cartridge and device (GeneXpert, Cepheid GmbH, 47807 Krefeld, Germany), as previously described [34,35]. Vitamin D metabolites were analyzed with a validated and strictly quality-controlled in-house LC-MS/MS method [33]. This method detects 25(OH)D_3_, 25(OH)D_2_, 24,25(OH)_2_D_3_, and 25,26(OH)_2_D_3_. Briefly, after pre-analytical sample preparation (50 µL serum), including protein precipitation (potassium hydroxide) and liquid/liquid extraction (n-heptane:tert-methyl-butyl-ether, 1 + 1) followed by derivatization with 4-phenyl-1,2,4-triazoline-3,5-dione (PTAD), samples were separated on a Nexera UHPLC from SHIMADZU (Kyoto, Japan) using a Kinetex^®^ 5 µm F5 100Å LC column (150 × 4.6 mm, Phenomenex, Torrance, CA, USA) with gradient elution. A SCIEX QTRAP 6500 triple quadrupole instrument (Applied Biosystems, Framingham, MA, USA) was employed for detection. This method has an LoQ of 3.1 nmol/L for 25(OH)D_3_, 1.0 nmol/L for 25(OH)D_2_, 24,25(OH)_2_D_3_, and 25,26(OH)_2_D_3_, respectively. Intra- and interassay imprecision is <9.6% across the entire analytical range. Total 25(OH)D is calculated by adding 25(O1H)D_3_ and 25(OH)D_2_. Interleukin-6 (IL-6) and C-reactive protein (CRP) were measured with commercial immunoassays on a COBAS 8000 analyzer (Roche Diagnostics, Rotkreuz, Switzerland).

### 2.3. Data Analysis

Median and interquartile ranges (IQR) were determined for not normally distributed variables. The analysis of categorical outcomes was performed by a Chi^2^ test. Continuous outcomes for independent samples were analyzed by the Mann–Whitney or Kruskal–Wallis test. For dependent continuous outcomes, Wilcoxon’s rank sum test was used. Associations between continuous outcomes were assessed by Spearman’s rank correlation. *p*-values < 0.05 were considered significant. The vitamin D metabolite ratio (VMR) was calculated as the ratio between 100 × 24,25(OH)_2_D_3_/25(OH)D (%). All statistical analyses were performed using SPSS statistical software (version 26.0; IBM Corp, Armonk, NY, USA).

## 3. Results

Baseline anthropometric and clinical characteristics are shown in Table 1. From the 148 patients included, 34 (23%) died within 30 days after admission. The frequency of fatal outcomes did not differ between males and females. Non-survivors were significantly older than survivors and had higher peak concentrations of IL-6 (*p* = 0.001) and CRP (*p* < 0.001) during hospitalization. Furthermore, non-survivors had a higher prevalence of renal (*p* = 0.001), and coronary artery disease (CAD, *p* = 0.028), and suffered more frequently from other pre-existing chronic diseases, including diabetes (*p* = 0.016) and cancer (*p* = 0.015). Interestingly, survivors had a significantly higher rate of hypertensive individuals than non-survivors (*p* = 0.006). Non-survivors required more frequent oxygen therapy and mechanical ventilation than non-survivors (*p* = 0.001). Nevertheless, the prevalence of antecedent pulmonary diseases did not differ significantly between both groups. Furthermore, 25(OH)D levels did not differ between the pulmonic treatment groups (*p* = 0.723, Kruskal–Wallis test, Figure 1). In COVID-19 non-survivors, the number of persons with vitamin D deficiency, defined by 25(OH)D serum levels below 30 nmol/L, did not differ significantly from that of survivors (*p* = 0.075).

The median serum concentration of 25(OH)D was 57.6 nmol/L. Sixty-one individuals had low 25(OH)D concentrations of less than 50 nmol/L. Table 2 provides the descriptive statistics of all measured vitamin D metabolites and the VMR. 24,25(OH)_2_D_3_, the principal catabolite of 25(OH)D_3_ accounted for 5.5% of the total 25(OH)D concentration. The second catabolite, 25,26(OH)_2_D_3_, was even less concentrated; 25(OH)D was highly inter-correlated with the two vitamin D catabolites 24,25(OH)_2_D_3_ (*r* = 0.884, *p* < 0.001) and 25,26(OH)_2_D_3_ (*r* = 0.932, *p* < 0.001).

Comparing the serum concentrations of all vitamin D metabolites and the VMR between survivors and non-survivors showed lower medians for 25(OH)D and VMR in non-survivors. Furthermore, the interquartile ranges of both parameters were substantially wider. However, these differences did not reach statistical significance. Additionally, the need for respiratory support was unrelated to the serum concentrations of 25(OH)D vitamin D and the two vitamin D catabolites, as well as the VMR.

## 4. Discussion

The present results do not show a significant association between serum indices of vitamin D metabolism and the 30-day outcome of hospitalized COVID-19 patients. The 25(OH)D, 24,25(OH)_2_D_3_, 25,26(OH)_2_D_3_ and the VMR were all unrelated to mortality and the need for ventilatory support. These findings extend existing knowledge not only by analysing 25(OH)D stores, but also the current status of vitamin D metabolism, which may be different in acutely diseased COVID-19 patients than in healthy individuals. Such differences are not captured by the simple measurement of 25(OH)D, which reflects the storage pool of inactive vitamin D.

Existing studies have reported consistently lower 25(OH)D serum concentrations in COVID-19 patients than in non-infected controls [36,37]. However, the clinical implications of lower 25(OH)D concentrations in COVID-19 patients are insufficiently understood. For example, Cereda et al. found no relationship between 25(OH)D (measured within 48 h after admission) and a wide range of clinical features. However, after adjusting for major confounders, they showed a positive association between 25(OH)D and in-hospital mortality, which is in contrast to the hypothesis that vitamin D deficiency increases the risk of Sars-CoV2 infection and an adverse clinical outcome [25]. Our results add to the growing body of evidence that does not support a relevant association between vitamin D metabolism and the severity or the course of COVID-19 [6,38,39,40]. For example, in a retrospective case–control study of 216 COVID-19 patients and 197 controls, Hernández et al. did not find any relationship between serum 25(OH)D and the severity of the disease [40]. Additionally, the risk of Sars-CoV2 infection is not consistently related to the 25(OH)D serum concentration [41]. For example, Al-Daghri et al. analyzed 138 patients with RT-PCR-confirmed Sars-CoV2 infection and 82 negative controls [41]. Although Sars-CoV2 patients had lower 25(OH)D concentrations than non-infected controls, Al-Daghri et al. could not show a higher risk of Sars-CoV2 infection in individuals with low 25(OH)D concentrations [41].

In contrast to the studies mentioned before, several others showed significant relationships between serum 25(OH)D and the Sars-CoV2 infection rate [42,43]. In addition, there are reports that support an association of 25(OH)D with COVID-19 severity [39,44,45] and mortality [19,40,46]. In order to reconcile the inconsistent results of existing studies, Akbar et al. performed a meta-analysis showing that low serum 25(OH)D concentrations are significantly related to the Sars-CoV2 infection rate and COVID-19 severity and mortality [47].

A general limitation of existing studies is that vitamin D status was assessed by the measurement of 25(OH)D, the inactive precursor of 1,25(OH)_2_D. However, the synthesis of active 1,25(OH)_2_D is typically maintained as long as 25(OH)D is available, and thus, serum 25(OH)D may not reliably indicate an impaired vitamin D metabolism. Furthermore, 25(OH)D measurements were mostly performed by different commercial immunoassays [28,48] that are known for their variable analytical performance [29,49], especially in critically ill patients [27,29]. Furthermore, cut-offs for vitamin D deficiency are primarily based on bone-related targets, such as bone mineral density and fracture risk. Whether or not these cut-offs also apply for other diseases is currently unclear [50].

The present study provides novel insights in the vitamin D metabolism of COVID-19 patients by the parallel measurement of 25(OH)D and the two catabolites 24,25(OH)_2_D_3_ and 25,26(OH)_2_D_3_. Our fully validated method provides accurate results for all metabolites and is not interfered by common matrix effects that are typically found in critically ill patients. Calculating the VMR allowed us to assess the current status of vitamin D metabolism. In the present cohort, median VMR was comparable to a cohort of non-diseased elderly individuals that we have analyzed previously [51]. Moreover, survivors and non-survivors had similar 24,25(OH)_2_D_3_ concentrations and VMR was not significantly different between these two groups. Additionally, the correlations between 25(OH)D, 24,25(OH)_2_D_3_, and 25,26(OH)_2_D_3_ were comparable in survivors and non-survivors. These results suggest that vitamin D metabolism is compromised neither in surviving nor in deceased COVID-19 patients. In contrast to previous studies, the parallel measurement of 25(OH)D, 24,25(OH)_2_D_3_, and 25,26(OH)_2_D_3_ facilitates an assessment of vitamin D metabolism that is based on dynamic physiologic principles and takes into account the individual set-point for vitamin D sufficiency. However, our cohort was rather well-supplied with vitamin D, as indicated by a median 25(OH)D concentration of 57.6 nmol/L. Consequently, the number of individuals with severe vitamin D deficiency was very low, which prevented a separate sub-group analysis of these patients. Hence, it cannot be excluded that pronounced vitamin D deficiency is related to the outcome of COVID-19 patients. Another possible explanation for the missing associations between vitamin D metabolite status and outcome in COVID-19 patients may be a rather weak immunomodulatory potential of vitamin D, which is insufficient to modulate the characteristic cytokine storm of COVID-19 patients, which goes along with massive cell destruction, endothelial cell deterioration, and hyper coagulation. Cereda et al. have discussed a dual role of vitamin D in COVID-19 [25]. On the one hand, it may boost mucosal defenses, and thus protect against infections; however, on the other hand, vitamin D seems to increase the expression of ACE2 receptors in many organs and on monocyte-derived macrophages [25]. An induction of ACE2 receptor expression might facilitate the entry of SARS-CoV-2 into host cells with subsequent macrophage activation and cytokine storm [25].

In our study, we observed a significant positive correlation between interleukin 6 and the vitamin D degradation products 24,25(OH)_2_D and 25,26(OH)_2_D (*p* = 0.002, Pearson correlation coefficient 0.37, respectively, not shown in results). This observation may indicate an enforced degradation of vitamin D in cases of high IL-6 levels due to enforced cytokine release. Nevertheless, the real cause is already not clear and needs further clarification. A substitution of vitamin D to compensate a possible degradation is probably too short-sighted because the degradation could also be a defense mechanism to avoid negative effects of vitamin D in seriously ill patients.

The inconsistent results of existing studies do not justify vitamin D supplementation in COVID-19 patients at the moment. A recent high-dose vitamin D supplementation study reported an improvement in disease severity and mortality [52]. In contrast, another placebo-controlled intervention study with a single dose of 200,000 IU of vitamin D3 did not significantly reduce the length of hospitalization [53]. The lack of consistent evidence for beneficial effects of vitamin D supplementation in COVID-19 patients was also emphasized in a recent editorial by Leaf and Ginde [54].

Limitations that should be considered when interpreting the results of our study include the moderate cohort size, a rather good supply with vitamin D, and the standardization of blood collection. Although many previous studies have analyzed smaller cohorts, a total of 148 participants and 34 non-survivors limits statistical power. The rather low number of individuals deficient in vitamin D further reduces the possibility of identifying potential associations and further subgroup analysis. Although we used the first serum sample that was sent to our laboratory, blood collections were usually not performed on the same day of hospitalization, but all were within the first 5 days after the patient’s admission to our center. However, 25(OH)D and other vitamin D metabolites have low biological variability and are not expected to change substantially within a few days. This fact is supported by the results from repeated blood collections of our patients that have been performed later. The vitamin D metabolite concentrations did not significantly differ from the initial values. On the other hand, our validated and strictly quality-controlled LC-MS/MS method ensures superior accuracy of the results and minimum interferences from matrix effects.

## 5. Conclusions

The present results do not support a relevant role of vitamin D for the course and outcome of COVID-19 in individuals that are not clearly deficient in vitamin D. Vitamin D metabolism, as indicated by the VMR, seems to be comparable in survivors and non-survivors, and is not related to the need of ventilatory support. Larger studies should confirm our results and address the impact of severe vitamin D deficiency on disease severity and fatal outcome in COVID-19 patients.

## Figures and Tables

**Figure 1 nutrients-13-02129-f001:**
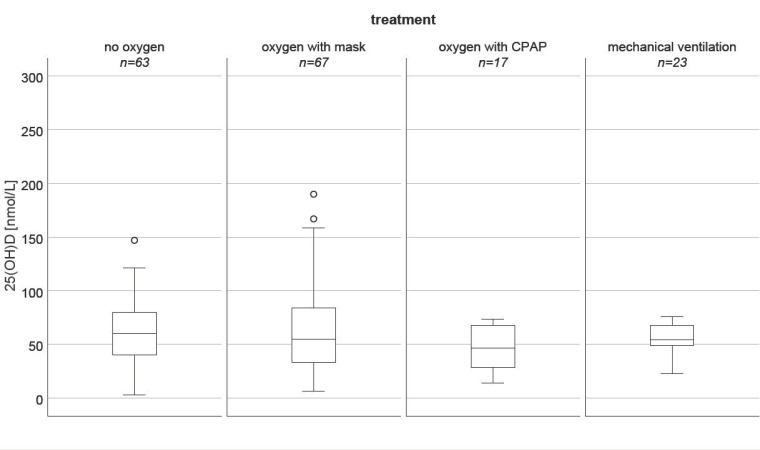
Difference of 25(OH)D (nmol/L) between the pulmonic treatment groups (*p* = 0.723, Kruskal–Wallis test).

**Table 1 nutrients-13-02129-t001:** Baseline characteristics of COVID-19 patients.

**Median, Q1–Q3**	**Exitus**		**Recovery**		***p*-Value ***
age	*n* = 34	80 (68–88)	*n* = 114	59 (45–78)	<0.001
peak IL-6 pg/mL during observation	*n* = 22	147.5 (89.2–279.0)	*n* = 55	47.1 (15.3–128)	0.001
peak CRP mg/L during observation	*n* = 32	103.2 (54.0–195.8)	*n* = 107	40.7 (6.5–107.6)	<0.001
**Total Number (%)**	**Exitus**		**Recovery**		***p*-Value ^+^**
female	15 (44.1)		56 (49.1)		0.608
male	19 (55.9)		58 (50.9)		
ICU admission	12 (35.3)		23 (20.2)		0.069
normal ward	22 (64.7)		91 (79.8)		
renal disease no	19 (55.9)		94 (82.5)		0.001
renal disease yes	15 (44.1)		20 (17.5)		
CAD no	19 (55.9)		86 (75.4)		0.028
CAD yes	15 (44.1)		28 (24.6)		
ambulant	0 (0)		35 (30.7)		<0.001
resident	34 (100)		79 (69.3)		
preexisting disease no	4 (11.8)		48 (42.1)		0.001
preexisting disease yes	30 (88.2)		66 (57.9)		
cancer no	24 (70.6)		103 (90.4)		0.015
cancer yes	9 (26.5)		10 (8.8)		
hypertension	10 (29.4)		64 (56.1)		0.006
normotension	24 (70.6)		50 (43.9)		
oxygen therapy no	4 (11.8)		50 (44.2)		0.001
oxygen therapy yes	18 (52.9)		44 (38.9)		
CPAP	3 (8.8)		10 (8.8)		
mechanical ventilation	9 (26.5)		9 (8.0)		
pulmonary disease no	28 (82.4)		98 (86.0)		0.701
pulmonary disease yes	6 (17.6)		15 (13.2)		
diabetes mellitus no	23 (67.6)		77 (86.5)		0.016
diabetes mellitus yes	11 (32.4)		12 (13.5)		
Vitamin D deficiency < 30 nmol/L yes	10 (29.4)		17 (14.9)		0.075
Vitamin D deficiency no	24 (70.6)		97 (85.1)		

* Mann–Whitney U test, ^+^ Chi2 test, CRP = C-reactive protein; CAD = coronary artery disease; CPAP = continuous positive airway pressure.

**Table 2 nutrients-13-02129-t002:** Vitamin D metabolite concentrations of hospitalized Sars-Cov-2 patients measured in the first routinely collected serum sample after admission.

	Total Cohort *n* = 148 (Analyzed)	Exitus *n* = 34	Recovery *n* = 114	*p*-Values * Exitus Versus Recovery
	Median (Q1–Q3)	Median (Q1–Q3)	Median (Q1–Q3)	
25(OH)D nmol/L	57.6 (34.3–81.5)	46.6 (22.8–89.1)	60.3 (37.3–81.0)	0.399 *
24,25(OH)_2_D_3_ nmol/L	3.1 (1.4–5.2)	3.2 (0.9–4.5)	3.1 (1.8–5.3)	0.192 *
25,26(OH)_2_D_3_ nmol/L	1.2 (0.7–1.7)	1.1 (0.5–1.9)	1.3 (0.8–1.7)	0.400 *
VMR [%]	5.5 (3.9–7.2)	4.4 (3.4–7.4)	5.5 (4.2–7.1)	0.308 *

Data are given in medians and interquartile ranges [(Q1) 25th–(Q3) 75th percentile]. A *p*-value < 0.05 was considered statistically significant. VMR [%] = Vitamin D metabolite ratio [24,25(OH)_2_D_3_/25(OH)D_3_ × 100]. * Mann–Whitney U test.

## Data Availability

The data are stored in a SPSS file at our institution accessible only by an authorized person.

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
