# Peer review of "Vitamin D Metabolites and Clinical Outcome in Hospitalized COVID-19 Patients"

_nutrients, 2021, doi:10.3390/nu13072129_

Round 1

Reviewer 1 Report

Since vitamin D can affect the immune system, it has been assumed that the vitamin D status may influence the prevalence and outcome of COVID-19. Despite several studies conducted in this field, data is still very heterogenous.  Zelzer et al. investigated if the circulating concentrations of vitamin D metabolites differ between survivors and non-survivors of hospitalized COIVD-19 patients. They found that the serum concentration of all vitamin D metabolites analysed did not differ significantly between the survivors and the non-survivors, and conclude that vitamin D does not play a relevant role in the coutcome of COVID-19.

This is a very intesting and well-conducted study that provided important data on the role of vitamin D in the the COVID-19 disease course.

The following comments should be addressed to improve the manuscript quality:

Title: Im my opinion the term „vitamin D metabolism“ is not correct, because the study analysed vitamin D metabolites that indicate the status, and not the metabolism. Thus, the title should be changed.

Abstract: At the beginning of the Abstract the authors note that the present study investigates the vitamin D metabolism in a mixed cohort of 148 hospitalized COVID-19 patients with potential fatal outcomes. However, vitamin D metabolism is more the measuring a couple of vitamin D metabolites. Thus, it would be more useful to mention that the present study aimed to investigate vitamin D status markers and vitamin D degradation products.

Introduction: At the end of the introduction, the origin and importance of 25,26(OH)2D3 should be explained. Currently, the authors provide only information on the metabolites 25(OH)D and 24,25(OH)2D.

Table 1: The abbreviations used in the Table (e.g. CAD, CPAP) should be defined in the Table legend.

Table 2: What do the numbers in brackets refer to? Is it %?

The 25(OH)D levels of the patients are comparatively high. I presume that only a minority of the patients had deficient 25(OH) D levels < 30 nmol/l. Please can you give information on the percentage of vitamin D-deficient patients using the classical cut-off levels in the fatal and non-fetal group.

Is there any relationship between vitamin D metabolites and the proinflammatory cytokines measured in the study (CRP, IL-6)? These data can maybe corroborate your conclusion.

Reviewer 2 Report

This is an interesting study examining the relationship between vitamin D and Covid19. Although there are now numerous studies looking at this relationship this study adds to our knowledge in two ways

1) The more epidemiological evidence the stronger the case - in this case for a lack of association at least in patients not frankly deficient 

2) This study looks at vitamin D metabolites - and uses LC Mass spec to measure the metabolites - the gold standard measurement method.

I have some questions/clarifications as follows

1) Can the authors clarify if this was a retrospective study? I.e. - was the blood analysed and "left over" originally drawn for some other reason and then the authors decided to analyse for vitamin D? Or was the blood drawn originally for this investigation - i.e prospective? Either way this should be specified

2) Line 13 of the abstract - what is meant by potentially fatal? do the authors mean ar risk of death?

3) Line 33- the two classic diseases of vitamin D deficiency are rickets and osteomalacia

4) Lines 35-36 - vitamin D has been associated with several diseases - I am not sure "is related to" is supported by the evidence - much of this evidence is associative, and intervention trials often disappointing

5) Lines 45-67 - the authors describe quite a few studies in this paragraph - please specify what types of studies - are they in vitro? in animal models? etc 

In particular, when discussing a meta-analysis please specify what type(s) of studies were analysed - observational? interventional? etc

6) Line 91- What do the authors mean by "validated"? Was a standard reference material used? This is the true gold standard when analysing vit D. Also, can the authors further explain/clarify the significance of the catabolic metabolites they measured? And the ratio they used

7) In terms of the data analysis section - did the authors have a hypothesis? Was this a secondary hypothesis? Did the authors do a post hoc study, and if so how did they correct for this? Why were two non-parametric tests (Mann-Whitney and Kruskal-Wallis) used, and how was it decided when each one was to be used?

8) Line 133- were the higher levels of IL6 and CRP physiologically relevant and by what percentage were they higher?

9) Line 152 - how was "p" determined? Spearman's rank typically gives an "r" but not a "p"

10) Any sub group analysis of frankly vitamin D deficient patients In relation to comments on line 263

Reviewer 3 Report

The present study investigates the vitamin D metabolism in a mixed cohort of 148 hospitalized COVID-19 patients with potential fatal outcomes.

The first problem that arises in this study is that the data from the direct control of the patients has not been collected. The data have been collected indirectly through laboratory and hospital information systems. Presumably, whoever provides these data does not know that they are going to be analyzed and therefore I consider this study to be of a similar quality to retrospective studies.

The time of collection of the blood sample is very imprecise: "the first routine blood collection that were sent to our laboratory for routine diagnostic". This could have been in the ER or within days of being hospitalized and without relation or knowledge of the patient's clinical status. We do not know if they were patients who were admitted with few symptoms but with risk factors such as cancer or the elderly, or patients with established pneumonia.

I also do not understand if the patients were asked for informed consent.

Vitamin D levels depend on many factors (obesity, institutionalization, skin color, substitution treatment, diet ...) that have not been taken into consideration in this study.

Reviewer 4 Report

This article is of interest in view of serum Vitamin D concentration in COVID-19 patients, exp. Exitus vs.

Recovery.

Mean 25OHD concentration was not different between the two groups, which is main points in this

study, however, the effect of this difference on mortality is not considered as clitical point.

Other factors such as age, underlying diseases severity should be considered as main

clitical points on the mortality not serum D levels.

In addition, the range of serum 25OHD concentration was not normal, both groups showed sub-optimal

level. It is not enough to compare to show the difference the two groups.

Round 2

Reviewer 1 Report

All critical points have been addressed by the authors

Reviewer 3 Report

The article has improved considerably and is very well written. But it has not been controlled whether or not patients were taking vitamin D replacement therapy. The low prevalence of vitamin D deficiency makes us suspect that there are many patients under treatment. Data with this bias are not very valid for drawing conclusions.

Reviewer 4 Report

Even though many limitations, this article have some informative results to consider.